# Exploring the Potential of the Model Cyanobacteria *Synechococcus* PCC 7002 and PCC 7942 for the Photoproduction of High-Value Terpenes: A Comparison with *Synechocystis* PCC 6803

**DOI:** 10.3390/biom13030504

**Published:** 2023-03-09

**Authors:** Célia Chenebault, Victoire Blanc-Garin, Marine Vincent, Encarnación Diaz-Santos, Amélie Goudet, Corinne Cassier-Chauvat, Franck Chauvat

**Affiliations:** 1Université Paris-Saclay, CEA, CNRS, Institute for Integrative Biology of the Cell (I2BC), 91198 Gif-sur-Yvette, France; 2CEA, INRAE, Département Médicaments et Technologies pour la Santé (DMTS), Université Paris Saclay, SCBM, 91191 Gif-sur-Yvette, France

**Keywords:** Cyanobacteria, bisabolene, farnesene, limonene, pinene, nitrogen sources (ammonium, nitrate, urea), genetic engineering, bioproduction

## Abstract

We have performed the first comparative analysis of the potential of two physiologically-diverse model cyanobacteria, *Synechococcus* PCC 7002 (S.7002) and *Synechococcus elongatus* PCC 7942 (S.7942), for the photosynthetic production of four chemically-different high-value terpenes: two monoterpenes limonene and pinene, and two sesquiterpenes bisabolene and farnesene. We showed, for the first time, that S.7002 and S.7942 can produce farnesene and bisabolene, respectively. Both cyanobacteria produced farnesene (S.7942 produced more efficiently than S.7002) more efficiently than the other tested terpenes (especially pinene, the weakest produced terpene). S.7002 produced limonene more efficiently than bisabolene, whereas S.7942 produced bisabolene more efficiently than limonene. These findings suggest that S.7942 is better suited to produce sesquiterpenes than monoterpenes. Interestingly, higher levels of terpenes were produced by S.7942 and S.7002 expressing a terpene-synthase gene from both an RSF1010-derived replicating plasmid and a neutral chromosomal site, as compared to either the plasmid alone or the chromosome alone. These results suggest that in both cyanobacteria, the production of terpenes is more limited by the activity of terpene synthases than the abundance of terpene precursors. Finally, higher levels of terpenes were produced by S.7002 growing on urea (a frequent pollutant) as compared to nitrate or ammonium, the standard nitrogen sources for cyanobacteria.

## 1. Introduction

Terpenes are a large family of chemicals with the formula (C_5_H_8_)_n_ that are often produced by plants where they act in their photoautotrophic metabolism, protection against pathogens, and/or attraction of pollinators. Many high-value terpenes are exploited by cosmetic, food, pharmaceutical, and/or biofuel industries, including bisabolene, farnesene, limonene, and pinene, the focus of the present study [1,2].

All terpenes are derived from the five-carbons (C5) building blocks, isopentenyl pyrophosphate (IPP), and dimethylallyl pyrophosphate (DMAPP) that are condensed to form the geranyl pyrophosphate (GPP, C10) precursor of monoterpenes (C_10_H_16_) such as limonene and pinene. The subsequent addition of one IPP unit on GPP forms the farnesyl pyrophosphate (FPP, C15) precursor of sesquiterpenes (C_15_H_24_), such as bisabolene and farnesene (Figure 1). 

Instead of plants that must be preserved for food production, cyanobacteria, the robust photosynthetic prokaryotes that colonize our planet, can be used for the sustainable production of terpenes from the plentiful natural resources: solar energy, water, CO_2,_ and minerals [1]. Cyanobacteria possess the methylerythritol-4-phosphate (MEP) pathway that produces GPP and FPP (Figure 1), which can be transformed into terpenes. For this purpose, synthetic genes encoding terpene synthases need to be adapted to the cyanobacterial codon usage and subsequently introduced and expressed in cyanobacteria [1,2].

The model (unicellular) cyanobacteria *Synechococcus* PCC 7002 (hereafter S.7002) and *Synechococcus elongatus* PCC 7942 (S.7942) are attractive for this purpose, in having good genetic tools and interesting physiological differences (for review, see [3]) which make their comparative analysis very interesting. The salt-tolerant marine model S.7002 (genome size 3.41 Mb) is able to grow on nitrate (the classical nitrogen source for cyanobacteria) or urea (cheaper than nitrate) [4], but it requires vitamin B12 for growing [5]. In contrast, the salt-sensitive freshwater model S.7942 (genome size 2.74 Mb) does not require vitamin B12 but is unable to grow on urea [6]. S.7002 and S.7942 have other key differences in their DNA repair [7] and glutathione-dependent systems that play a prominent role in tolerance to stresses [3].

Two proof-of-concept studies showed that S.7002 can be engineered for the photosynthetic production of bisabolene and limonene [8] or pinene [9] but not farnesene, as we presently report. Concerning S.7942, several laboratories reported that S.7942 can be engineered for the production of farnesene [10,11,12] or limonene [13,14], but neither bisabolene nor pinene, as we report here. The limited number of previous studies and their multiple experimental differences (nature of the terpene synthase genes and their promoters, and of the chromosome/plasmid vectors propagating them) make it difficult to know whether S.7002 and/or S.7942 are suitable chassis for the photoproduction of chemically-diverse terpenes or only a few of them.

In this study, we have performed the first comparative analysis of the potential of the two physiologically-diverse model cyanobacteria, S.7002 and S.7942, for the photosynthetic production of four chemically-different high-value terpenes. Practically, we have employed the same genetic strategy to engineer S.7002 and S.7942 for the photoproduction of two monoterpenes, limonene (cyclic molecule) and pinene (bicyclic), and two sesquiterpenes, bisabolene (cyclic) and farnesene (linear), in order to rationally compare the potential of these two model cyanobacteria for the production of diverse terpenes. Each terpene synthase gene was expressed from the same strong promoter propagated by the same autonomously replicating plasmid vector in both S.7002 and S.7942. We showed for the first time that S.7002 and S.7942 can produce farnesene and bisabolene, respectively. Both S.7002 and S.7942 produced farnesene and pinene with the best (farnesene) and the lowest (pinene) efficiencies. Furthermore, S.7002 produced limonene better than bisabolene, whereas S.7942 produced bisabolene more efficiently than limonene. We also showed for the first time that S.7002 cultivated on urea (a frequent pollutant) produces terpenes more efficiently than cultures growing on either nitrate or ammonium (standard but more expensive nitrogen sources for cyanobacteria). Finally, we showed in both S.7002 and S.7942 that increasing the copy number of the terpene synthase-encoding gene increases the level of terpene production, suggesting that terpene production is more limited by the activity of terpene synthases rather than the abundance of terpene precursors. 

## 2. Materials and Methods

### 2.1. Bacterial Strains and Growth Conditions

*E. coli* strains used for gene manipulations (TOP10 and NEB10 beta, Appendix A) or conjugative transfer of pC-derived replicative plasmids (Appendix A) to S.7942 or S.7002 (CM404 [15]) were grown at 37 °C (TOP10 and NEB10 beta) or 30 °C (CM404) on LB medium containing appropriate antibiotics: ampicillin (Ap) 100 μg·mL^−1^, kanamycin (Km) 50 μg·mL^−1^, streptomycin (Sm) 25 μg·mL^−1^ or spectinomycin (Sp) 75 μg·mL^−1^.

S.7942 and S.7002 were grown at 30 °C, under continuous white light (2500 lux; 31.25 μE·m^−2^·s^−1^) and agitation (140 rpm, Infors rotary shaker) in liquid mineral medium: MM, i.e., BG11 [16] enriched with 3.78 mM Na_2_CO_3_ [17] for S.7942, or A+ containing B12 vitamin (4 μg·L^−1^) for S.7002 [18]. The terpene-producing strains were grown in the presence of the selective antibiotics: Km 50 μg·mL^−1^ for both S.7002 and S.7942, and/or Sm 50 μg·mL^−1^ and Sp 50 μg·mL^−1^ for S.7002 and Sm 5 μg·mL^−1^ and Sp 5 μg·mL^−1^ for S.7942. For some experiments carried out with S.7002, the standard nitrogen source (NaNO_3_, 12 mM) was replaced by urea or ammonium. Nitrate-grown cells were washed twice and resuspended in nitrogen-free media that were supplemented with either urea or nitrate, as indicated. Growth in liquid cultures was monitored by regular measurements of optical density at 750 nm (OD_750_) using a spectrophotometer (Jenway 6700). 

### 2.2. Genetic Manipulations

The pC-derived replicative plasmids for high-level expression of terpene synthase genes (Appendix A) were introduced in S.7942 and S.7002 by conjugation, as described [19,20], using a 72 h co-incubation of *E. coli* and cyanobacterial cells. The DNA cassettes for high-level expression of terpene synthase genes were introduced in neutral sites of the chromosome of S.7942 or S.7002 by transformation, as described [21,22]. Cells were then plated on MM containing Km 50 μg·mL^−1^ or both Sp 5 μg·mL^−1^ and Sm 5 μg·mL^−1^ (S.7942), or A+ medium containing 4 μg·L^−1^ B12 vitamin, 3 g·L^−1^ sodium thiosulfate, Km 50 μg·mL^−1^ or both Sp 50 μg·mL^−1^ and Sm 50 μg·mL^−1^ (S.7002), which were solidified with 1% (S.7942) or 1.5% (S.7002) Bacto Agar (Difco). 

The presence of the terpene-synthase-encoding DNA cassette propagated in pC-derived plasmids or a neutral chromosomal site was verified by PCR and DNA sequencing (Mix2Seq Kit, Eurofins Genomics) using appropriate oligonucleotide primers (Appendix A).

### 2.3. Terpene Collection and Quantification by Gas Chromatography-Mass Spectrometry

S.7002 and S.7942 strains engineered for terpene production were photoautotrophically grown in the presence of selective antibiotics in 250 mL erlenmeyers containing 50 mL cell suspensions overlaid with 20% (vol/vol) dodecane (analytical grade, Sigma-Aldrich) to trap terpenes [1,20,23]. At time intervals, 300 μL aliquots of the dodecane overlay were collected, and 1 μL aliquots of these dodecane samples were injected in a split mode 10:1 (limonene) or 5:1 (E-α-bisabolene, farnesene, and pinene) into a GC-MS apparatus (Trace1300 (GC) + ISQ LT (MS), ThermoScientific). The quantification of terpenes was performed as we previously described [20,23]. 

## 3. Results and Discussion

### 3.1. Construction of the Synechococcus PCC 7002 and the Synechococcus Elongatus PCC 7942 Strains Expressing the Studied Terpene Synthase Genes from the Strong Lambda-phage pR Promoter and Propagated in Either or Both a Neutral Chromosomal Site or a Replicative Plasmid

In this study, we tested and compared the ability of the unicellular model cyanobacteria S.7002 and S.7942 to produce the four chemically-different high-value terpenes: bisabolene, farnesene, limonene, and pinene (Figure 1). For this purpose, we used the following terpene synthases-encoding genes that worked well in cyanobacteria [1], namely: (i) the *Abies grandis* E-α-bisabolene synthase (bs), (ii) the *Picea abies* α-farnesene synthase (fs), (iii) the *Mentha spicata* 4S-limonene synthase gene (ls) and (iv) the *Pinus taeda* (+)-α-pinene synthase gene (ps). Each codon-adapted terpene synthase gene was cloned downstream of the strong lambda-phage p*R*-promoter and associated cro ribosome-binding site of the autonomously replicating RSF1010-derived [15,19] pC plasmid vector [24], yielding the plasmids pCBS, pCFS, pCLS, and pCPS [20,23]. These plasmids were introduced by conjugation in S.7002 and S.7942 [15,23], yielding the strains represented in Figure 2. In each case, two independent Sm^R^/Sp^R^ clones were selected and analyzed by PCR and DNA sequencing (Appendix A) to verify that pCBS, pCFS, pCLS, and pCPS plasmids replicate stably in S.7002 and S.7942, without affecting their photoautotrophic growth (Appendix A).

The pR-LS and pR-FS recombinant genes were also cloned in the NS_7002_ and NSI_7942_ neutral chromosomal sites of S.7002 and S.7942, respectively, to compare the level of terpene production driven by either or both the plasmid and the chromosome (Figure 2).

### 3.2. Unlike Synechococcus PCC 7002, the Growth of Synechococcus elongatus PCC 7942 Is Affected by the Dodecane Overlay Used to Trap Terpenes, but Dodecane-Adapted Strains Grow Healthy under Dodecane

As terpenes are volatile chemicals, cyanobacteria engineered for terpene production are routinely grown under a dodecane overlay to trap terpenes, which are subsequently quantified by gas chromatography coupled with mass spectrometry [1]. The growth of S.7002 was not affected by a dodecane overlay (Figure 3A), in agreement with previous studies on S.7002 [8,9]. In contrast, the growth of S.7942 was strongly delayed by a dodecane overlay (Figure 3B), but S.7942 strains precultivated under dodecane grew subsequently well under dodecane (Figure 3), a finding not discussed by previous workers [10,11,12]. Consequently, we used dodecane-adapted strains to analyze terpene production by S.7942.

### 3.3. The First Report of Farnesene Production by Synechococcus PCC 7002: The Production Is Lower Than That of Synechococcus elongatus PCC 7942

The production of farnesene during the photoautotrophic growth of the S.7002 and S.7942 strains propagating the pCFS plasmid, or the empty pC plasmid vector used as a negative control, were measured over 21-day periods of photoautotrophic growths. GC-MS analysis of dodecane overlay samples from the pCFS propagating strains showed a peak with a similar retention time and ion chromatogram than a pure standard of α-farnesene, which was not observed in the negative-control pC strains (Appendix A). These data showed that both S.7002 and S.7942 can photosynthetically produce farnesene (Figure 4A), a novel finding in the case of S.7002. However, lower levels of farnesene were produced by S.7002 (~2 mg·L^−1^ after 21 days, Figure 4), as compared to S.7942 (~8 mg·L^−1^ after 21 days, Figure 4A). 

The level of farnesene production by S.7942 (~1.5 mg·L^−1^ after seven days, Figure 4A) was roughly similar to what was reported by another group (~3 mg·L^−1^ after eight days, no subsequent measurements) [10,12]) who cloned into a neutral chromosomal site (we used a replicative plasmid) a codon-adapted α-farnesene synthase gene from *Malus domestica Borkh* or *Picea abies* (we used only the *Picea abies* enzyme) expressed from the strong p*Trc* promoter (we used the strong lambda-phage p*R* promoter). However, the multiple experimental differences between these reports and our own study make their comparison difficult. This explains why we employed the same genetic engineering strategy to explore the capability of various model cyanobacteria to photosynthetically produce various high-value terpenes. 

The level of farnesene production driven by the pCFS plasmid in S.7942 and S.7002 could be really compared to the level driven by the same pCFS plasmid we previously reported in *Synechocystis* PCC 6803 (hereafter S.6803) [20], another unicellular model cyanobacterium phylogenetically-distant to both S.7942 and S7002 [3]. Interestingly, the pCFS-driven farnesene production in S.7942 and S.7002 (~8 and ~2 mg·L^−1^ after 21 days, respectively, Figure 4A) was lower than what we observed in S.6803 harboring the same pCFS plasmid (~30 mg·L^−1^ after 21 days, [20], Appendix A). These findings indicate that S.7942 and S.7002 are less suitable cyanobacterial factories than S.6803 for the photoproduction of farnesene. 

### 3.4. First Report of Bisabolene Production by Synechococcus Elongatus PCC 7942: The Production Is Similar to That of Synechococcus PCC 7002

The photosynthetic production of bisabolene by the S.7002 and S.7942 strains propagating the pCBS plasmid, or the empty pC plasmid, was measured over a 21-day period of cultures (Figure 4B). GC-MS analysis of dodecane overlay samples from the pCBS strains showed a peak with a similar retention time (Appendix A) and ion chromatogram than a pure standard of E-α-bisabolene, which was not observed in the negative-control pC strain (Appendix A). Both S.7002 and S.7942 produced similar levels of bisabolene (~0.5 and ~2.0 mg·L^−1^ after 21 days, respectively, Figure 4B).

The production of bisabolene by S.7002 (~0.1 mg·L^−1^ after four days; ~0.5 mg·L^−1^ after 21 days) is lower than the previously reported level (~0.5 mg·L^−1^ after six days, no subsequent measurement [8]) observed after cloning the *Abies grandis* (E)-α-bisabolene synthase gene expressed from the *cpc* phycocyanin promoter (we used the lambda-phage p*R* promoter) into a neutral chromosomal site (we used a replicative plasmid). The important differences between the previous report and our study make their comparison difficult.

The presently-observed levels of bisabolene production driven by the pCBS plasmid in S.7002 and S.7942 (~0.5 and ~2.0 mg·L^−1^ after 21 days, respectively, as seen in Figure 4B) were lower than the value we previously reported for S.6803 harboring the same pCBS plasmid (10–14 mg·L^−1^ after 21 days, [20,25]). Together with the above-mentioned findings, these data (Appendix A) indicate that S.7942 and S.7002 are less efficient cyanobacterial factories than S.6803 for the photoproduction of the sesquiterpenes farnesene and bisabolene.

### 3.5. The Synechococcus PCC 7002 and Synechococcus elongatus PCC 7942 Strains Harboring the pCLS Plasmid Produce Limonene in This Order

The photosynthetic production of limonene by the S.7002 and S.7942 strains propagating the pCLS plasmid, or the empty pC plasmid, was monitored for 21 days (Figure 4C). Dodecane overlay samples from the pCLS strains, not the pC strains, showed a GC-MS peak typical of a pure S-(-)-limonene standard (Appendix A). Interestingly, a higher level of limonene was produced by S.7002 (~1.0 mg·L^−1^ after 14 days and ~1.7 mg·L^−1^ after 21 days, Figure 4C) as compared to S.7942 (~0.3 mg·L^−1^ after 14 days, Figure 4C). 

The presently-observed production of limonene by S.7002 (~0.3 mg·L^−1^ after four days, Figure 4C) is lower than the previously reported level (~4.0 mg·L^−1^ after four days, but no subsequent measurement [8]) observed after cloning into a neutral chromosomal site (we used a replicative plasmid) the *Mentha spicata* limonene synthase gene expressed from the *cpc* phycocyanin promoter (we used the lambda-phage p*R* promoter). 

Similarly, the presently-observed production of limonene by S.7942 (~0.06 mg·L^−1^ after seven days, Figure 4C) seems lower than the level (~17 mg·L^−1^ after seven days, but no subsequent measurement [14]) observed after cloning into a neutral chromosomal site (we used a replicative plasmid) the *Mentha spicata* limonene synthase gene expressed from the *psbA* promoter (we used the lambda-phage p*R* promoter). 

The multiple experimental differences between the previous reports and our study make it difficult to ascribe the difference in limonene production to one or several parameters: growth conditions, nature of the promoter expressing the limonene synthase gene, and/or of the cloning vehicle propagating it.

The presently-observed level of limonene production driven by the pCLS plasmid in S.7002 (~1.7 mg·L^−1^ after 21 days, Figure 4C) is slightly higher than the value we previously observed in S.6803 (~0.7 mg·L^−1^ after 21 days [20,25]. These findings (Appendix A) indicate that S.7002 is a cyanobacterial cell factory as effective as S.6803 for the production of the monoterpene limonene, unlike what we observed in the case of the two sesquiterpenes: bisabolene and farnesene (Appendix A).

### 3.6. The Synechococcus PCC 7002 and Synechococcus elongatus PCC 7942 Strains Harboring the pCPS Plasmid Produce Pinene Inefficiently

The photosynthetic production of pinene by the S.7002 and S.7942 strains propagating the pCPS plasmid, or the empty pC plasmid, was measured for 21 days (Figure 4D). GC-MS analysis of dodecane overlay samples from the pCPS strain of S.7002 showed a peak typical of a pure standard of α-pinene, which was not observed in the pC strain (Appendix A). In contrast, no pinene production was detected in S.7942 harboring pCPS (Figure 4D).

The level of pinene production by S.7002 (~60 μg·L^−1^ after 21 days, Figure 4D) is similar to what we previously reported for S.6803 harboring the same pCPS plasmid (~80 μg·L^−1^ after 21 days; [20], Appendix A).

The presently observed production of pinene by S.7002 (~60 μg·L^−1^ after 21 days) is lower than the previously reported level (1.5 mg·L^−1^ after six days, [9]) observed after cloning into the endogenous pAQ1 plasmid (we used the non-cyanobacterial pC vector derived from the broad-host-range plasmid RSF1010 [20]) a *Abies grandis* gene encoding a 6xhistidine-tagged pinene synthase (we used the native *Pinus taeda* enzyme) expressed from the *cpc* promoter (we used the lambda-phage p*R* promoter). As both the pAQ1 and pC vectors, on the one hand, and the *cpc* and p*R* promoters, on the other hand, function well in cyanobacteria [3], we assume that the 6xhistidine-tagged pinene synthase from *Abies grandis* is more efficient than the native *Pinus taeda* enzyme we have used in this study (Figure 4D), and our previous analysis of pinene production in S.6803 [20], Appendix A).

### 3.7. Comparison of Limonene Production in Synechococcus PCC 7002 Driven by the Recombinant LS Gene Propagated by Either or Both the pCLS Replicative Plasmid and the NS_7002_ Neutral Chromosomal Site

Because the activity of an enzyme can be improved by increasing the copy number of the gene encoding it, we attempted to increase the limonene production of the S.7002 strain harboring pCLS by cloning an extra copy of the same p*R*-LS recombinant gene (LS expressed from the strong p*R* promoter) in the NS_7002_ neutral chromosomal site (a region in between A0935-A0936 [26]) often used for gene cloning in S.7002 [3]. Therefore, a p*R*-LS-Km^R^ DNA cassette was assembled by Gibson [27] (Appendix A) in between the two 500-bp DNA regions surrounding NS_7002_ that serve as a platform of homology for DNA recombination promoting targeted integration of the p*R*-LS-Km^R^ cartridge into the chromosome. The resulting pTwist_NS-S7002p_R_LSKm^R^ plasmid (Appendix A) was transformed to S.7002 pCLS cells, where the p*R*-LS-Km^R^ DNA cassette was integrated into all chromosome copies (Appendix A). The resulting strain expressing the p*R*-LS gene from both the pCLS plasmid and the chromosome was designated as pCLS + chrLS (Figure 2). 

In parallel, the pTwist_NS-S7002p_R_LSKm^R^ plasmid was transformed to S.7002 wild-type cells, where the p*R*-LS-Km^R^ DNA cassette was integrated into all chromosome copies (Appendix A). The resulting strain named chrLS served to measure the level of limonene production (Figure 2) driven by the chromosomal p*R*-LS gene.

All three strains, pCLS, chrLS, and pCLS + chrLS, grew well under standard photoautotrophic conditions in the presence of the dodecane overlay (Figure 5). Interestingly, the level of limonene produced by the pCLS + chrLS strain was roughly equivalent to the sum of limonene produced by the strains pCLS and chrLS, which were similarly active (Figure 5). These results indicate that the production of terpenes by an engineered S.7002 strain is probably more limited by the activity of the employed terpene synthase than the abundance of terpene precursors, as we recently discussed in the case of S.6803 [20].

### 3.8. Comparison of Farnesene Production in Synechococcus Elongatus PCC 7942 Driven by the Recombinant FS Gene Propagated by Either or Both the pCFS Replicative Plasmid and the NSI_7942_ Neutral Chromosomal Site 

We also attempted to increase farnesene production in the pCFS S.7942 strain by cloning an extra copy of the p*R*-FS recombinant gene (FS expressed from the p*R* promoter) in the NSI_7942_ neutral chromosomal site (GenBank accession U30252, [28]) a site often used for gene cloning in S.7942 [3]. Therefore, a p*R*-FS-Km^R^ DNA cassette was assembled by Gibson [27] (Appendix A) in between the two 300-bp DNA regions surrounding NSI_7942_ to mediate the targeted integration of the p*R*-FS-Km^R^ cartridge in NSI_7942_. The resulting pTwist_NSI-S7942 p_R_FSKm^R^ plasmid (Appendix A) was transformed to S.7942 pCFS cells where the p*R*-FS-Km^R^ DNA cassette was integrated into the NSI_7942_ site of all chromosome copies (Appendix A). The resulting strain expressing the p*R*-FS gene from both the pCFS plasmid and the chromosome was designated as pCFS + chrFS (Figure 2). 

In parallel, the pTwist_NSI-S7942 p_R_FSKm^R^ plasmid was transformed to S.7942 wild-type cells where the p*R*-FS-Km^R^ DNA cassette was integrated into the NSI_7942_ site of all chromosome copies (Appendix A). The resulting strain named chrFS (Figure 2) served to measure the level of farnesene production driven by the chromosomal p*R*-FS gene (Figure 6).

The two pCFS and chrFS grew slightly better than the pCFS + chrFS strain under standard photoautotrophic conditions, irrespective of the presence of the terpene-trapping dodecane overlay (Figure 6A). Interestingly, the level of farnesene produced by the pCFS + chrFS strain was roughly equivalent to the sum of farnesene produced by the strains pFCS and chrFS, which were similarly active (Figure 6B). These results indicate that the production of terpenes by an engineered S.7942 strain is probably more limited by the terpene-synthase activity than the terpene-precursors abundance, as we recently discussed in the case of S.6803 [20]. This assumption is consistent with the finding that terpene synthases are slow enzymes and that selecting a terpene synthase with high Kcat and low Km is pivotal in terpenoid biosynthesis [29].

### 3.9. Influence of Growth Conditions on Terpene Production: First Report That Synechococcus PCC 7002 Cultivated on Urea Produces Terpenes More Efficiently Than Cultures Growing on Nitrate or Ammonium in This Order 

In the view of reducing the costs of future industrial production of terpenes, eventually coupled with wastewater treatment, we have tested the influence of urea (frequently present in natural waters [4]) on cell growth and the production of the monoterpene limonene and the sesquiterpene farnesene. For this purpose, we have used S.7002 because it grows well on all three nitrogen sources [5], ammonium, nitrate (the standard nitrogen source for cyanobacteria [6]), and urea (cheaper than nitrate), unlike S.7942, which cannot grow on urea because it has neither urea transport (UrtABCDE) nor catabolic (UreABCDEF) enzymes [4]. In these assays, S.7002 strains harboring the pFS or pCLS plasmids were cultivated in the same standard concentration of nitrogen (12 mM) supplied as either nitrate (NaNO_3_, 12 mM), ammonium (NH_4_Cl, 12 mM) or urea (CO(NH_2_)_2_, 6 mM). In the case of urea (and nitrate as a control), we tested both protocols in which nitrogen is supplied either all at once, at the beginning of the culture, or provided gradually as successive doses during cultivation as was required by the other cyanobacteria *Cyanothece* PCC 7425 and *Synechocystis* PCC 6803 [20,23]. The results (Figure 7) showed, for the first time, that S.7002 growing on urea as the sole nitrogen source can produce terpene more efficiently than cells cultivated on nitrate or ammonium (in this order). Interestingly, *Cyanothece* PCC 7425 and S.6803 produced terpene less efficiently (C.7425) or similarly (S.6803) when cultivated on urea as compared to nitrate [20,23].

## 4. Conclusions

We have performed the first rational and comparative analysis of the ability of two physiologically-diverse model unicellular cyanobacteria, *Synechococcus* PCC 7002 (a marine host) and *Synechococcus elongatus* PCC 7942 (a freshwater host), to photosynthetically produce chemically-diverse terpenes of high-value (bisabolene, farnesene, limonene, and pinene). For this purpose, each terpene synthase gene was expressed from the same strong promoter and propagated by the same RSF1010-derived replicative plasmid vector. We showed, for the first time, that S.7002 and S.7942 can be engineered for the photoproduction of farnesene and bisabolene, respectively. S.7942 and S.7002 (in this order) produced farnesene more efficiently than the other tested terpenes, especially pinene, the weakest produced terpene, similar to what we previously observed in the other model cyanobacterium *Synechocystis* PCC 6803 (S.6803) [20]. These findings indicate that the *Pinus taeda* pinene synthase does not work well in cyanobacteria. Furthermore, S.7002 produced limonene more efficiently than bisabolene, whereas S.7942 produced bisabolene more efficiently than limonene. We showed for the first time that S.7002 can produce terpenes when growing on urea (a frequent pollutant) as the sole nitrogen source, similar to what we observed for S.6803 [20]. In the case of S.7002, higher levels of terpenes were produced by cultures growing on urea, as compared to nitrate or ammonium, the standard or frequent nitrogen source for cyanobacteria, respectively. By contrast, neither S.7942 nor the fastest growing strain, *Synechococcus elongatus* UTEX 2973, can be used for chemical production from urea because they both lack all urea transport and catabolism enzymes [4]. Interestingly, higher levels of terpenes were produced by both S.7002 and S.7942 expressing the terpene synthase gene from both an RSF1010-derived replicating pC plasmid and a neutral chromosomal site, as we observed in *Synechocystis* PCC 6803 [20]. These results suggest that the production of terpenes in cyanobacteria is more limited by the activity of terpene synthases than the abundance of terpene precursors, similar to what we discussed in the case of S.6803 [20]. Finally, we compared the levels of terpene production by S.7002 and S.7942 with what we previously observed for S.6803, which was engineered in the same way as S.7002 and S.7942 [20]. We found that S.6803 produces higher levels of the sesquiterpenes farnesene and bisabolene than S7942 and S.7002 (in this order), suggesting that S.6803 could be a better cyanobacterial chassis for future industrial photoproduction of these chemicals.

## Figures and Tables

**Figure 1 biomolecules-13-00504-f001:**
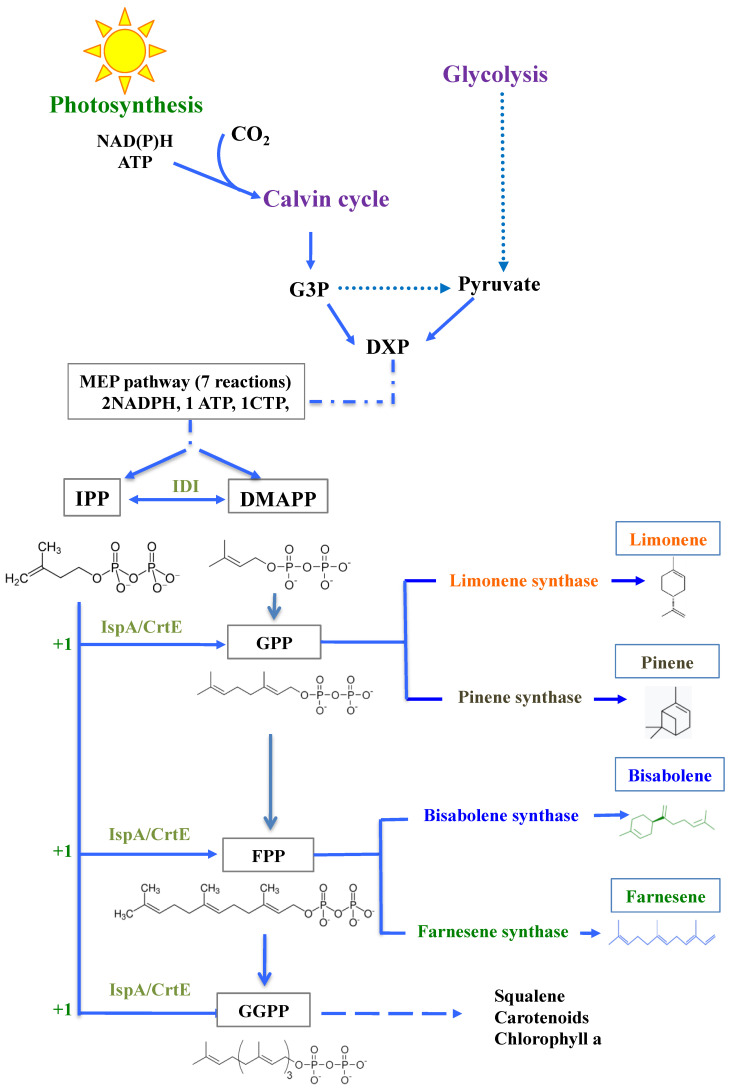
Schematic representation of the metabolic pathway and key compounds acting in the synthesis of terpenes from CO_2_. Abbreviations: DMAPP: dimethylallyl pyrophosphate; DXP: 1-deoxy-d-xylulose-5-phosphate; FPP: farnesyl pyrophosphate; GPP: geranyl pyrophosphate; GGPP: geranylgeranyl pyrophosphate; G3P: glyceraldehyde 3 phosphate; IPP: isopentenyl pyrophosphate; IDI: isopentenyl-diphosphate isomerase; IspA/CrtE: isoprenyl diphosphate synthase; MEP: methylerythritol 4-phosphate (it consumes 2 NADPH, 1 ATP, and 1 CTP).

**Figure 2 biomolecules-13-00504-f002:**
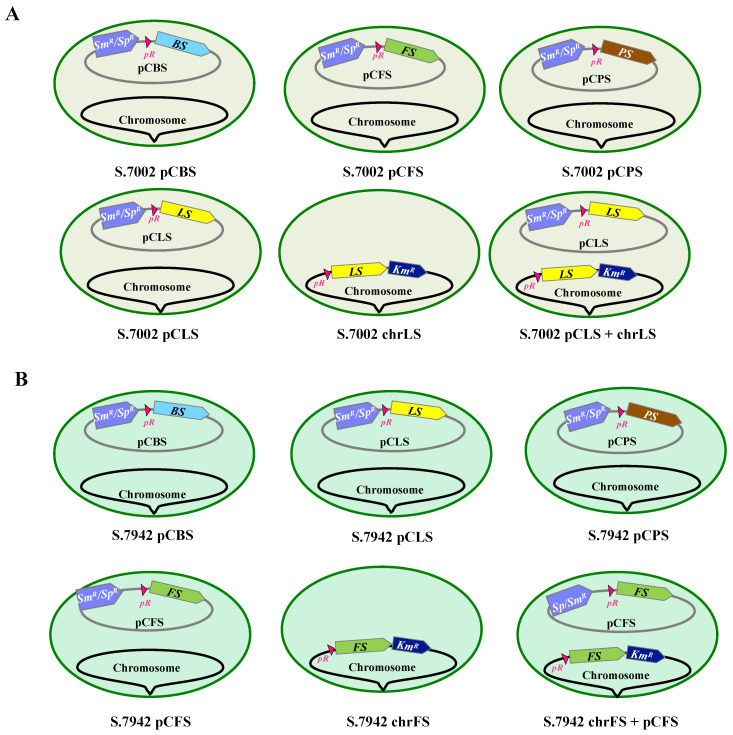
Schematic representation of the cyanobacterial strains engineered in this study. S.7002 (**A**) and S.7942 (**B**) cells are represented as green ovals, while their chromosome (black phylactery line) is arbitrarily shown as attached to the cell membrane to distinguish it from the pC-derived replicative plasmids. These Sm^R^/Sp^R^ plasmids, which express a terpene-synthase encoding gene from the strong λ phage *pR* promoter (red triangle), are named pCBS (bisabolene synthase, light-blue arrow), pCFS (farnesene synthase, green arrow), pCLS (limonene synthase, yellow arrow) and pCPS (pinene synthase, brown arrow). Note that the chromosome of two strains of both S.7002 and S.7942 harbor a *pR*-LS (Km^R^) and a *pR*-FS (Km^R^) gene cassette, respectively. They served for the comparison of terpene production driven by either or both the replicating plasmid and the chromosome.

**Figure 3 biomolecules-13-00504-f003:**
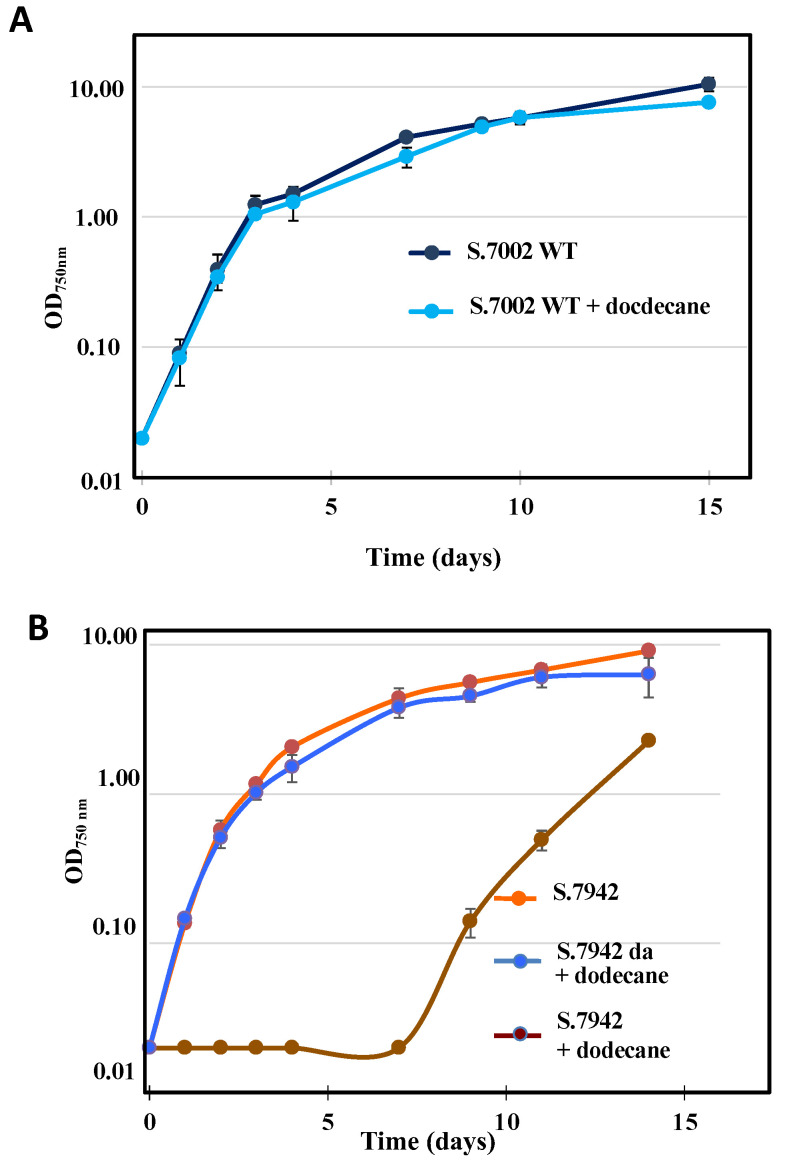
Influence of a dodecane overlay on the photoautotrophic growth of Synechococcus PCC 7002 (S.7002; (**A**)) and Synechococcus elongatus PCC 7942 (S.7942; (**B**)). Cells were grown photoautotrophically in the absence or presence of a dodecane (20% vol/vol) overlay. S.7942, denoted as da (da for dodecane-adapted cultures), was inoculated with cells that were precultivated under dodecane for more than two weeks. Error bars represent standard deviations from biological triplicates.

**Figure 4 biomolecules-13-00504-f004:**
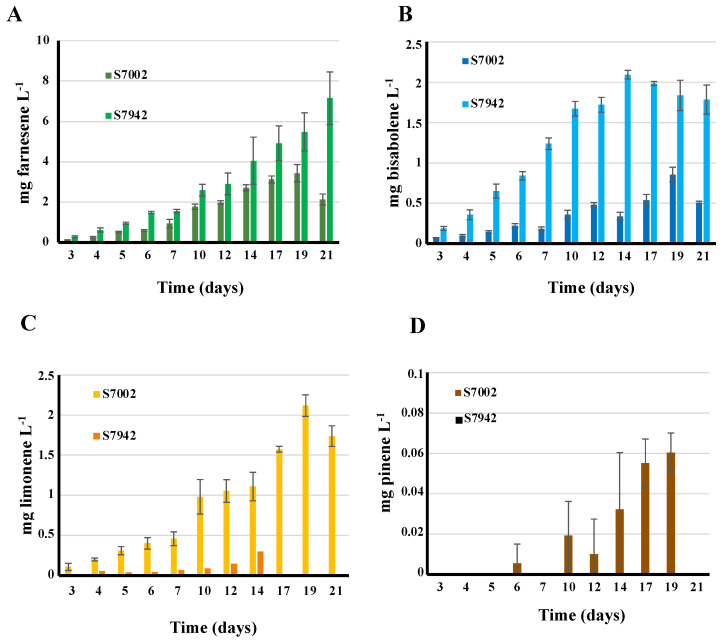
Comparison of the levels of terpene production by the *Synechococcus* PCC 7002 and *Synechococcus elongatus* PCC 7942 strains expressing a terpene synthase gene from the following plasmids: (**A**) pCFS (farnesene synthase), (**B**) pCBS (bisabolene synthase), (**C**) pCLS (limonene synthase) and (**D**) pCPS (pinene synthase). Cells were grown under dodecane (20% vol/vol) in otherwise standard photoautotrophic conditions to assay terpene production for 21 days. Error bars represent standard deviation from *n* ≥ 2 biological replicates. Note that note that no pinene was produced by S.7942.

**Figure 5 biomolecules-13-00504-f005:**
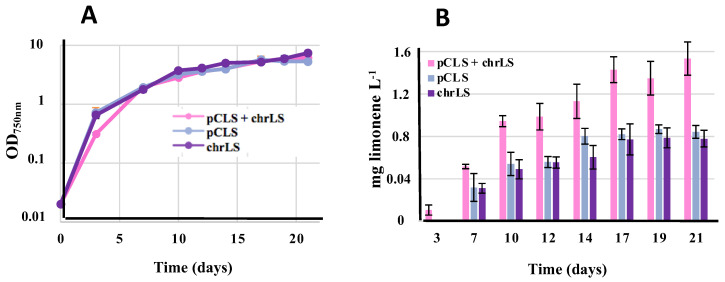
Simultaneous analysis of growth (**A**) and terpene production (**B**) of the *Synechococcus* PCC 7002 strains harboring the strongly expressed limonene synthase gene (LS) in either or both the replative plasmid (pCLS) or a neutral chromosomal site (chrLS). Cells were grown under standard photoautotrophic conditions in the presence of a dodecane overlay (20% vol/vol) to assay terpene production for 21 days. Error bars represent standard deviation from biological triplicates.

**Figure 6 biomolecules-13-00504-f006:**
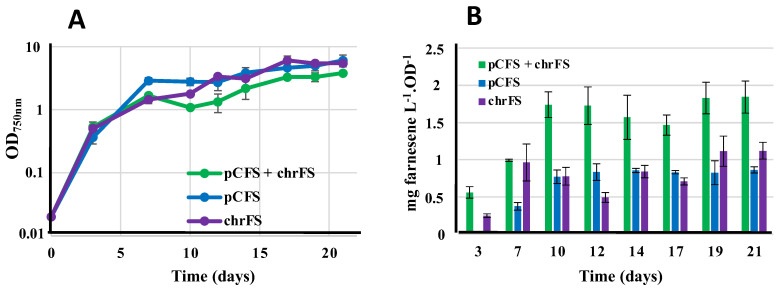
Simultaneous analysis of growth (**A**) and terpene production (**B**) of the *Synechococcus elongatus* PCC 7942 strains harboring the strongly expressed farnesene synthase gene (FS) in either or both the replative plasmid (pCFS) or a neutral chromosomal site (chrFS). Cells were grown under standard photoautotrophic conditions in the presence of a dodecane overlay (20% vol/vol) to assay terpene production for 21 days. Error bars represent standard deviation from biological triplicates.

**Figure 7 biomolecules-13-00504-f007:**
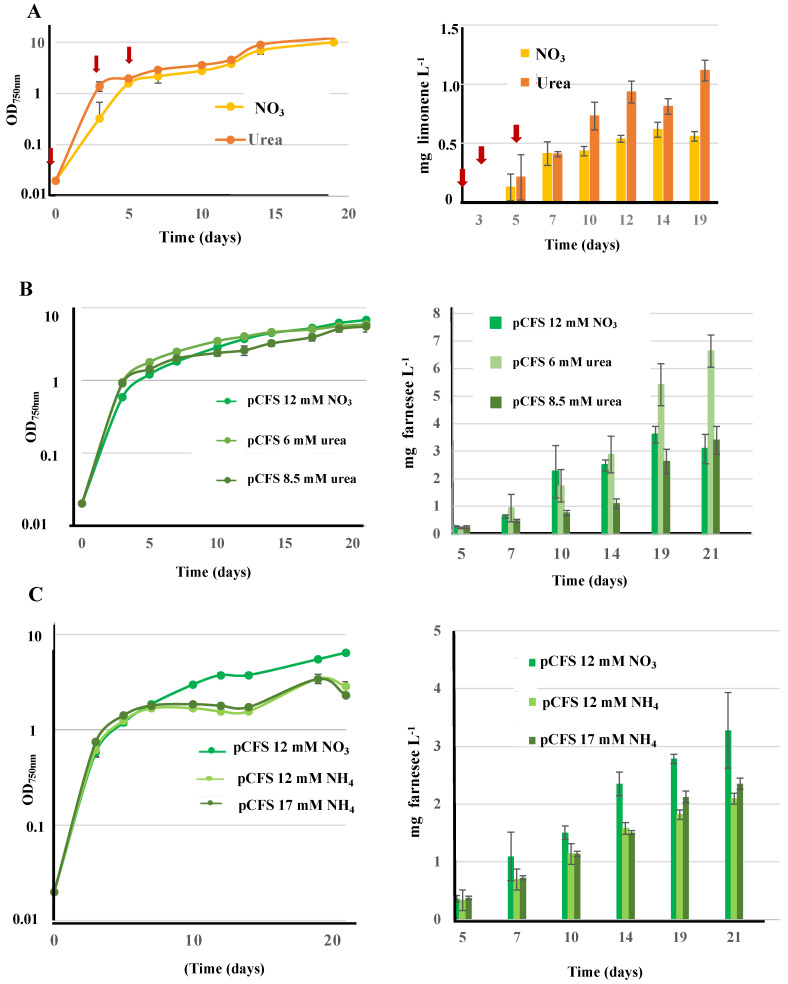
Influence of various nitrogen sources on the growth and terpene production of the *Synechococcus* PCC 7002 strains harboring pCLS (**A**) or pCFS (**B**,**C**). Cells growing exponentially under standard conditions (12 mM nitrate) were washed twice and resuspended in a nitrogen-free medium. (**A**) At the times indicated by red arrows (days 0, 3, and 5), the same amount of nitrogen was supplied as either 4 mM NaNO_3_ or 2 mM urea. (**B**,**C**) The indicated doses of either nitrate or urea (**B**), ammonium (NH_4_Cl), or nitrate (**C**) were supplied at the onset of the cultures. In all of these assays, the dodecane overlay (20% vol/vol) was added on day 3. Error bars represent standard deviations from biological triplicates.

## Data Availability

Not applicable.

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
