# Peer review of "Exploring the Potential of the Model Cyanobacteria Synechococcus PCC 7002 and PCC 7942 for the Photoproduction of High-Value Terpenes: A Comparison with Synechocystis PCC 6803"

_biomolecules, 2023, doi:10.3390/biom13030504_

Round 1
Reviewer 1 Report
The manuscript (biomolecules-2239682) is relevant to the scientific community, mainly to add more knowledge to the area of production of high-value chemicals, which is important for the development of biotechnological products. However, a few changes are recommended for subsequent publication.
Appointments:
1. The English language is good, but a little spell check is required. Example: in the line 388 (caption of Figure 7) - "… under standzrd …".
2. Figure 1 - Figure needs improvement. Some of the structures need to be redrawn, sharpening, removing shading or contours. The abbreviation GPP must be included and defined in the caption.
3. The scientific names must be carefully corrected throughout the text. Several species or strains cited are not written in italics or are missing "sp". Example of some corrections that should be made:
a. Line 144 and 145 – “…Abies grandis …(ii) the Picea abies …, (iii) the Mentha spicata … and (iv) the Pinus taeda …”
b. And the Lines 187, 202, 203, 211, 216, 220 and 221 (caption of Figure 4), 226, 237, 248, 291, 292, 326 and others.
Reviewer 2 Report
In this manuscript, the authors engineer two strains of cyanobacteria, PCC 7002 and PCC 7942, for the production of mono- and sesquiterpenes. The experiments seem to be well designed and well executed and will be of interest as a starting point for further engineering efforts in those species.
The research is not particularly novel, as it has been shown many times that transforming terpene synthases into various organisms will cause the organisms to produce small amounts of terpene products. Also, the observation that the terpene synthase itself is a major bottleneck in terpene production has been reported many times by other researchers (it's a little weird that the authors only cite themselves on this point). For me, the most interesting result in the paper was the result showing higher terpene production by PCC 7002 in urea than other nitrogen sources. Also, the result of improved production from adapting the PCC 7002 to dodecane was interesting.
Many of the authors conclusions relate to which species they think would be more suited for production of various terpenes. I think most of these kinds of conclusions are premature, as terpene production levels from all the species is still far below what would be commercially viable, and many optimization techniques remain to be explored, such as media and growth conditions, precursor pathway engineering, and terpene synthase selection/expression optimization.
Some typos:
"abondance" should be changed to "abundance" throughout.
line 46: preserved --> reserved
line 93: "IDI: isopentenyl-diphosphate;" --> "IDI: isopentenyl-diphosphate isomerase;"
line 200: "an other" --> "another"
line 380: "whashed" --> "washed"
Figure 7 is kind of a mess and is difficult to understand. In particular: some of the words have red underlines. The x-axis on the bottom left figure has a typo. The bottom right figure has no y-axis label. The legend of the bottom right figure seems cutoff. Middle-right figure, the legend blocks some of the data and error bars. I don't understand why the pCFS 12 mM NO3 looks so different in B vs C, I suppose at the 21 day time point it is similar in both experiments.
Supplemental figures:
Some of them have those red "spellcheck" underlines under various words. Figures should be remade without those.
For the mass-spec figures the left panels are described as "Ion chromatograms". Are they Total ion chromatograms or extracted ion chromatograms? If they are extracted ions, what m/z is it? Also, the methods don't say anything about the GC conditions or temperature program, it would be nice to have those listed in this manuscript instead of relying on a reference.
Reviewer 3 Report
The manuscript can be accepted after revision.
1. Actually, the exact name for "Synechococcus PCC 7942" should be "Synechococcus elongatus PCC 7942". Thus, the related section in the title and the text should be revised when it first appears.
2. Modify the scale of Y-axis in Figure 3 as it's hard to see the exact OD for each day.
3. It's better to compare the yield (mg/g) but not production (mg/L) between S7002 and S7942 to offset the influence of different biomass.
4. L211, "Synechocystis" should be italic. (also Line 216, same problems in legends of Figure 4, please check out other sentences.); the abbreviation of S6803 in Line 216 should be first defined in L211.
Line 215-216 and also similar conclusions, this conclusion could only make sense in the authors' tested conditions. The authors only express the related genes with a promoter without optimizing the pathways or carbon flux. The production potential for different strains could be different as production of farnesene in S7942 has reached 1.2 mg/L/Day after optimization which may not be surely realized in S6803. Thus, the authors should limit their conclusions in specific conditions.
